

# A roadmap for empowering cardiovascular disease patients: a 5P-Medicine approach and technological integration

Hanna V. Denysyuk[1], Ivan Miguel Pires[2] and Nuno M. Garcia[3]

[1] Instituto de Telecomunicações, Universidade da Beira Interior, Covilhã, Portugal
[2] Instituto de Telecomunicações, Escola Superior de Tecnologia e Gestão de Águeda, Universidade de Aveiro, Águeda, Portugal
[3] Instituto de Biofísica e Engenharia Biomédica, Faculdade de Ciências, Universidade de Lisboa, Lisboa, Portugal

Corresponding author
Hanna V. Denysyuk,
hanna.denysyuk@ubi.pt

## ABSTRACT

This article explores the multifaceted concept of cardiovascular disease (CVD) patients' empowerment, emphasizing a shift from compliance-oriented models to active patient participation. In recognizing that cardiovascular disease is a paramount global health challenge, this study illuminates the pressing need for empowering patients, underscoring their role as active participants in their healthcare journey. Grounded in 5P-Medicine principles—Predictive, Preventive, Participatory, Personalized, and Precision Medicine—the importance of empowering CVD patients through analytics, prevention, participatory decision making, and personalized treatments is highlighted. Incorporating a comprehensive overview of patient empowerment strategies, including self-management, health literacy, patient involvement, and shared decision making, the article advocates for tailored approaches aligned with individual needs, cultural contexts, and healthcare systems. Technological integration is examined to enhance patient engagement and personalized healthcare experiences. The critical role of patient-centered design in integrating digital tools for CVD management is emphasized, ensuring successful adoption and meaningful impact on healthcare outcomes. The conclusion proposes vital research questions addressing challenges and opportunities in CVD patient empowerment. These questions stress the importance of medical community research, understanding user expectations, evaluating existing technologies, defining ideal empowerment scenarios, and conducting a literature review for informed advancements. This article lays the foundation for future research, contributing to ongoing patient-centered healthcare evolution, especially in empowering individuals with a 5P-Medicine approach to cardiovascular diseases.

## INTRODUCTION

Disorders of the heart and blood vessels, such as peripheral artery disease, heart failure, stroke, and coronary artery disease, are referred to as cardiovascular disease (CVD) (*Lopez, Ballard & Jan, 2022*; *World Health Organization, 2008a*). CVD is a severe global health

concern and is responsible for a considerable burden of sickness and mortality worldwide (*Roth et al., 2020*).

Among the world's major causes of death is cardiovascular disease. The World Health Organization (WHO) estimates that cardiovascular disease (CVD) kills 17.9 million people a year, or roughly 31% of all fatalities worldwide (*World Health Organization, 2008a*). People of various ages and backgrounds are impacted by it. Nonetheless, the frequency rises with age and among lower socioeconomic level population groups, members of racial and ethnic minorities, and members of marginalized communities (*Suglia et al., 2020*; *Williams et al., 2010*; *Shepherd et al., 2018*). CVD-related health disparities emphasize how critical it is to address social determinants of health and advance fair access to healthcare services and preventive measures (*Mannoh et al., 2021*; *Teshale et al., 2023*; *Powell-Wiley et al., 2022*).

Chronic non-communicable illnesses (NCDs), such as cancer, diabetes, and chronic respiratory conditions, are categorized with CVDs (*World Health Organization, 2011*; *Boutayeb, Boutayeb & Boutayeb, 2013*; *World Health Organization, 2008a*; *Kuruvilla, Mishra & Ghosh, 2023*). The main goals of the fight against cardiovascular disease are risk factor management, early identification, and prevention (*Faghy et al., 2023*; *World Health Organization, 2008a*, *2008c*).

Normalizing blood pressure, lowering stress, eating a balanced diet, putting smoking cessation strategies into practice, expanding access to high-quality healthcare, and doing research to further our understanding of CVD and its treatment choices are all examples of public health activities (*Tang et al., 2017*; *Okorare et al., 2023*; *Bays et al., 2021*).

One of the most prevalent chronic illnesses worldwide, CVD is associated with significant societal and financial expenses as well as high rates of morbidity, mortality, and loss of quality of life (*Roth et al., 2020*; *Amini, Zayeri & Salehi, 2021*). Establishing a strategy for empowering cardiovascular patients has become imperative due to the rising number of deaths caused by CVDs. The primary obstacle confronting healthcare establishments, such as clinics and hospitals, is the availability of dependable facilities at reasonable costs (*Roth et al., 2020*; *Williams, Walker & Egede, 2016*; *Kruk et al., 2018*).

Enhancing communication between patients and healthcare providers, guaranteeing the effective use of primary health resources, and promoting adherence to treatment regimens are all made possible by patient empowerment. The World Health Organization's Regional Office for Europe outlines it as a specific aim in Health 2020 (*Wallerstein, 2006*). People are increasingly recognized as co-producers of their health and need to be empowered to take control of the determinants of their health, according to this policy framework, which offers the primary strategies and priorities to promote European action for health and well-being (*Wallerstein, 2006*; *Braveman & Gottlieb, 2014*).

Our previous work (*Pires et al., 2021*) focus on the conceptual foundation and initial proposal of a novel system architecture that integrates the 5P-Medicine paradigm into the management of cardiovascular diseases. The key contributions of this article include: i) a detailed conceptual framework for the 5P-Medicine approach, highlighting how each aspect (Predictive, Preventive, Participatory, Personalized, and Precision) can be integrated into a cohesive system for cardiovascular patients; ii) provides a comprehensive

review of the state-of-the-art in mobile and wearable personal devices used in digital health care, specifically for cardiovascular diseases; iii) proposes a detailed system architecture that includes data acquisition, cloud-based processing, and patient-healthcare provider interaction; iv) discusses the challenges related to implementing this system, including data acquisition, sensor integration, and real-world applicability. The study's limitations are also clearly identified, particularly that the proposed system still needs to be implemented or tested in a real-world setting.

The primary limitation was the conceptual nature of the study, with the proposed system still needing to be implemented. The analysis was theoretical, and practical implications could only be assessed after implementation and pilot testing in multiple countries.

The current article builds upon the previous conceptual framework and delves into practical implementation aspects. This article emphasizes the development of a comprehensive patient empowerment framework, detailing how empowerment methodologies can be effectively integrated into the 5P-Medicine approach. It explores integrating digital health tools, mobile applications, wearable devices, and telemedicine to enhance patient engagement and self-management. The article also outlines critical questions and future research directions for finetuning a roadmap for empowering cardiovascular disease (CVD) patients using ICT technologies. It proposes questions related to the acceptance of wearables, expectations from the medical and patient communities, and barriers to technology adoption. This work provides a meta-vision for developing solutions that empower CVD patients, establishing a detailed roadmap that aligns with the 5P-Medicine vision, and addressing previous limitations by focusing on real-world applications, including experimental challenges and integration with national health systems.

In this article, the authors comprehensively compare patient empowerment (PE) methodologies across various health conditions, demonstrating how the 5P-Medicine framework significantly enhances PE strategies. The impact of 5P-Medicine on PE is profound, as it incorporates and interconnects the Predictive, Preventive, Participatory, Personalized, and Precision aspects of healthcare. Integrating these five facets, the 5P-Medicine model positions patient empowerment at its core, fostering an environment where patients are actively engaged in their health management. This interconnected approach ensures that predictive algorithms can foresee health issues, preventive measures can be tailored to individual needs, participatory strategies encourage patient involvement, personalized plans are developed based on unique patient profiles, and precision treatments are administered with high accuracy. Collectively, these elements work synergistically to enhance patient autonomy, improve health outcomes, and transform the traditional healthcare paradigm into a more patient-centric model.

In summary, the current article explicitly continues and substantively expands upon the first article's work. It transitions from establishing a theoretical framework for 5P-Medicine in cardiovascular care to addressing practical implementation strategies for patient empowerment. This narrative delineates the unique contributions of each article while demonstrating their interconnectedness as part of an ongoing research project.

This article aims to outline a possible roadmap that enables the research, design, and development of an ecosystem that supports the empowerment of CVD patients while using Information and Communication Technologies (ICT).

The motivation for this article is based on several assumptions:

1. We assume that CVD patients are specific regarding their requirements for empowerment when compared with other health conditions that are currently ahead in the patient empowerment roadmap;

2. The community still has to agree on what it means to be an empowered CVD patient; without at least a proposal on what it means to be an empowered CVD patient, there will be little progress towards this main goal;

3. That the empowerment of CVD patients connects perfectly to a vision of 5P-Medicine;

4. The design of new medical devices must be guided in view of the requirements of a specific group of users/patients to overcome/facilitate the adoption of these devices.

Additionally, several questions must be addressed before finetuning a roadmap for empowering CVD patients. In particular, and assuming a 5P-Medicine vision, we propose the following questions:

1. What wearables and smart devices are accepted by the medical community as support devices to diagnose/monitor/support therapeutics and disease management?

2. What are the expectations of the patient community and the medical community regarding the empowerment of CVD patients?

3. Are these expectations met to this day? If yes, how and to what extent; if not, why? What are the technological barriers?

4. What would be the ideal scenario for CVD patient empowerment using wearables and smart devices?

5. What is the most relevant literature on solutions that allow diagnosis/monitoring/ support to therapeutics and support to disease management and self-management for CVD patients?

Yet, answering these questions is outside the scope of this article and will be addressed later in the research.

Therefore, the main purpose of this article is to establish a meta-vision of what could be a roadmap for developing solutions that empower a CVD patient.

Furthermore, this article aims to define this vision by integrating principles from 5P-Medicine into patient empowerment methodologies. The 5P-Medicine approach typically encompasses personalized, predictive, preventive, participatory, and precision aspects of healthcare by leveraging emerging technologies such as digital health tools, mobile applications, wearable devices, telemedicine, and other innovative solutions that can enhance patient engagement, self-management, and communication with healthcare providers.

Pursuing of transforming CVD management, integrating the 5P-Medicine approach offers a comprehensive framework that emphasizes patient empowerment. The 5P-Medicine paradigm-Predictive, Preventive, Participatory, Personalized, and Precision Medicine-provides a multi-faceted strategy to enhance patient care and engagement. The following roadmap, represented in Fig. 1, illustrates how these principles can be practically applied to empower CVD patients, ensuring they become proactive participants in their healthcare journey. The Predictive aspect focuses on early detection and proactive management of cardiovascular risks through advanced diagnostics and continuous monitoring using wearable technology. Preventive Medicine aims to reduce cardiovascular disease risk through lifestyle changes, regular screenings, and vaccinations. Participatory Medicine emphasizes involving patients in decision making, enhancing their health literacy, and empowering them to manage their health actively. Personalized Medicine tailors treatments based on individual genetic profiles and health data, ensuring better patient outcomes and satisfaction. Finally, Precision Medicine utilizes advanced diagnostics and targeted therapies to administer precise treatments, minimizing side effects and improving the effectiveness of disease management. This holistic approach integrates technological advancements and patient-centred care to empower individuals in managing their cardiovascular health.

The main contributions revolve around the conceptualization of patient empowerment, its application to cardiovascular disease management, the integration of digital tools, alignment with 5P-Medicine, and the importance of user-centered design in medical device development. The article advocates for a patient-centered approach to healthcare that prioritizes individual needs, preferences, and active participation in decision making for improved health outcomes.

The review primarily benefits CVD patients by promoting empowerment through self-management, health literacy, and participatory decision making, enhancing their ability to manage their health. Healthcare professionals gain insights into the advantages of patient-centered care, leading to better treatment adherence and satisfaction. Healthcare systems and policymakers benefit from more efficient resource use and improved population health outcomes. Researchers and medical device developers are guided to create user-centered technologies tailored to patient needs, driving innovation and acceptance. Finally, it advocates for a holistic, patient-centered approach to CVD management, benefiting all stakeholders.

# SEARCH METHODOLOGY

## Literature search strategy

This article proposes a novel methodology that synergizes advanced technology with patient-centric healthcare strategies. The aim is to develop a framework that facilitates efficient healthcare delivery and empowers patients in their healthcare journey. The search methodology follows informal research representing an embedded study grounded in interdisciplinary approaches, combining insights from medicine, technology, and patient psychology. This work provides a theoretical framework for future study and helps to develop a more nuanced understanding of the research problem.

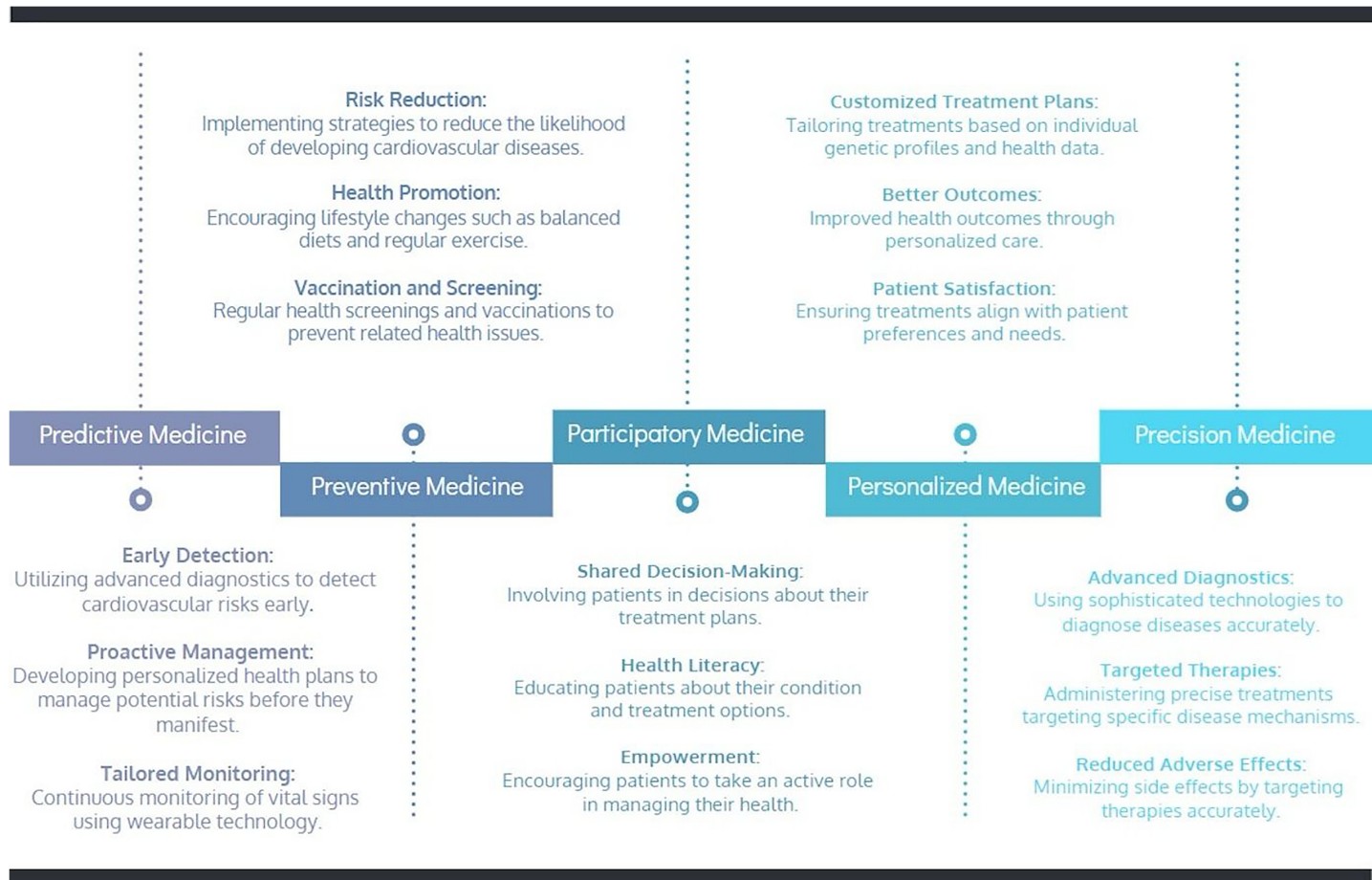

**Figure 1 Roadmap for empowering cardiovascular disease patients: 5P-Medicine approach.**

Explicit inclusion and exclusion criteria were established to focus on relevant literature from diverse sources, including Springer, Elsevier, MDPI, JMIR, IEEE, PLOS ONE, Google Scholar, and other online sources. A systematic and exhaustive search strategy was developed to cover various keywords and phrases related to patient empowerment, cardiovascular disease, digital health, and relevant technologies. It ensured that literature from various sub-disciplines and perspectives was considered.

The literature review underwent a rigorous peer review process to validate the inclusion of studies and ensure the quality and credibility of selected sources. The research team included professionals with diverse backgrounds, expertise, and perspectives. This diversity contributed to a more balanced interpretation of the literature, reducing the risk of biases associated with a singular viewpoint.

The literature review process was iterative, allowing continuous refinement based on emerging findings and insights.

## Data extraction and synthesis

The study extracted data from selected studies using a standardized form, including study design, population characteristics, intervention details, outcome measures, and key

findings. Quality assessment was done by assessing adaptability and mapping different categories. Authors independently rated each study, and discrepancies were resolved through discussion or consultation. A narrative synthesis summarized the findings.

## Sample size determination

In this literature review, the rationale for the sample sizes used in the analyzed studies is critically evaluated to ensure the robustness and reliability of the findings. Adequate sample sizes are crucial for detecting significant effects and making valid inferences about the effectiveness of patient empowerment strategies in CVD management.

Several studies analyzed in this review conducted power analyses to determine the minimum sample size required to detect significant effects with adequate power. These guidelines helped ensure that the studies were sufficiently powered to detect small to significant effects of patient empowerment interventions on outcomes such as self-management behaviors, health literacy, and patient satisfaction.

Overall, the sample sizes in the analyzed studies were generally adequate to detect significant effects of patient empowerment strategies. Studies that followed rigorous power analysis procedures and provided detailed sample size justifications were more likely to report reliable and valid findings. This review underscores the importance of appropriate sample size determination in research on patient empowerment interventions to ensure that the reported effects are statistically and practically significant.

## Statistical analysis

The study utilized descriptive analysis to summarize study characteristics, meta-analysis techniques, and conduct subgroup analyses to identify potential heterogeneity sources, such as differences in study design or population characteristics, to ensure robustness and reliability in the extracted data.

## PATIENT EMPOWERMENT

"Empowerment" has been recognized in the healthcare industry as a substitute for compliance in directing the provider-patient interaction. The empowerment-oriented approach to health care sees patients as accountable for their decisions and results, unlike the more conventional compliance-oriented approach, which sees patients as the recipients of medical decisions and prescriptions. *Gibson (1991)* made an effort to define the concept. However, there are numerous interpretations of the phrase "empowerment" based on various understandings of the idea (*Laverack & Wallerstein, 2001*). It is challenging to think of empowerment consistently and in ways that make it feasible for its application in healthcare settings, as the analysis reveals that the notion is linked to several qualities. According to some researchers, empowerment can mean different things to different people in different situations (*Rappapon, 1984*). On the other hand, the authors of (*Kieffer, 1984*) contend that a clear and practical definition is required to demonstrate the usefulness of the term "empowerment" for theory and practice.

The overall goal of empowerment in the healthcare industry is to replace the conventional hierarchical relationship between patients and clinicians with a more patient-

centered, collaborative one. The empowerment process gives individuals, such as patients, healthcare professionals, and communities, the ability to take charge of their health, make educated decisions, and actively participate in their medical care. It entails giving people the information, abilities, tools, and encouragement to speak up for their health needs and choices. It helps people to take an active role in their care, make educated decisions, and enhance their general well-being and health results.

Initially, patient empowerment (PE) was proposed to promote health (*World Health Organization, 1986*). It came about as a result of the realization that giving patients medical advice and treatments alone would not be adequate to produce the best possible health results.

Patient empowerment is a complex process of personal transformation that healthcare providers may assist. It is based on the idea that everyone has the innate ability to be accountable for their own lives (*Werbrouck et al., 2018*; *Aujoulat, d'Hoore & Deccache, 2007*; *Funnell, 2004*). According to several academics, patient empowerment includes encouraging self-management (*Bravo et al., 2015*).

There are various ways to describe patient empowerment based on the goals and environment. Patient empowerment is defined by the European Patient Forum (EPF) as giving patients the knowledge and ability to take charge of their healthcare and well-being by actively participating in their treatment, making educated decisions, and having their preferences honored (*European Patients Forum, 2017*). The American Heart Association defines patient empowerment as the capacity of patients to comprehend their health status and available treatments, effectively interact with healthcare providers, actively engage in joint decision making, and assume accountability for their health (*Virani et al., 2020*). According to the Institute of Medicine, patient empowerment is the process by which people are given more authority over choices and behaviors that impact their health (*Kelly & Fuster, 2010*). The European Network for Patient Empowerment (*Enope, 2023*) states that an empowered activated patient is aware of their health condition and how it affects their body, feels capable of participating in decision making with their healthcare providers, is aware that lifestyle modifications are required to manage their condition, can question and challenge the providers of their care, assumes responsibility for their health and actively seeks care when needed, and actively looks for, evaluates, and uses information. Furthermore, patient empowerment is promoted by the U.S. government's Agency for Healthcare Research and Quality (AHRQ) as a means of assisting individuals in gaining the knowledge, abilities, attitudes, and self-awareness necessary to manage their health and healthcare properly (*Sinsky & Krist, 2010*).

These definitions emphasize the fundamental ideas of patient empowerment, such as shared decision making, self-management, information access, active participation in healthcare decisions, and the capacity to take charge of one's health.

Participatory health informatics (PHI) examines how technology can help people make better decisions and manage their health by enhancing health literacy and the doctor-patient bond. It allows patients to participate more actively in their care (*Denecke et al., 2021*; *Vahdat et al., 2014*).

Global health depends heavily on patient empowerment (*Wakefield et al., 2018*), and the degree to which international patient empowerment initiatives are put into practice varies based on the resources and healthcare system of the specific nation. According to the WHO, patient empowerment is a dynamic process in which patients thoroughly understand their roles and responsibilities. Healthcare providers equip them with the requisite information and skills, allowing them to carry out healthcare-related jobs efficiently. This process occurs in a setting that values and respects cultural and community diversity. In addition, patients are actively urged to take an active role in their care by making knowledgeable decisions and having meaningful conversations with medical staff (*World Health Organization, 2008c*). According to recent data, high-income nations, especially those in Europe, are at the forefront of this field (*Bombard et al., 2018*).

## Patient empowerment strategies

In the past few decades, patient empowerment has been more popular as a proactive collaboration and patient-centered self-care approach to enhance the quality of life and health outcomes for those with chronic illnesses (*Neuhauser, 2003*; *Rosenfield, 1992*). Support groups, self-monitoring, tailored information, feedback, educational opportunities, personal exercise programs, communication (*e.g.*, chat, email) with patients or health care providers, patient decision making, changes in health care services, and advocacy efforts are examples of empowerment interventions that have been actively pursued in the following areas: diabetes care and other chronic diseases (*Howorka et al., 2000*; *Lorig, Ritter & González, 2003*; *Lorig et al., 2001*; *Mayer-Davis et al., 2004*; *Cooper, Booth & Gill, 2003*), chronic obstructive pulmonary disease (*Endicott et al., 2003*), end-stage renal disease (*Tsay & Hung, 2004*), osteoporosis (*Groessl, 1999*), disabilities (*Bhagwanjee & Stewart, 1999*), cancer (*Golant, Altman & Martin, 2003*; *Mishra et al., 1998*; *Davison & Degner, 1997*). Two systematic reviews have demonstrated the benefits of self-management education for patients with diabetes, where eleven group-based education studies have reported improvements in diabetes control, knowledge, and medication need, which are linked to higher levels of treatment satisfaction, self-empowerment, and self-management skills (*Deakin et al., 2005*). With empowering features like patient decision making and small group discussion more effective than didactic sessions, seventy-two studies demonstrated short-term impacts in self-management, dietary habits, and illness control (*Norris, Engelgau & Venkat Narayan, 2001*).

Family empowerment techniques have improved caregiver efficacy, coping mechanisms, access, efficient use of health resources, and individual patient empowerment. The most critical applications of family strategies in mental health have been reported (*Sherman, 2003*; *Dixon et al., 2001*); these include lower anxiety and depression when caring for children with chronic illnesses (*Melnyk et al., 2004*). Reduced sadness, anxiety, and increased empowerment were observed in support group interventions including grandparents and in a systematic evaluation of 20 parent training trials aimed at improving the psychosocial health of mothers (*McCallion, Janicki & Kolomer, 2004*; *Barlow, Coren & Stewart-Brown, 2003*).

The evidence presented in *World Health Organization (2008b)* indicates that patient empowerment strategies affect health outcomes in several ways: directly, through improvements in the effectiveness of individual decision making, management of disease complications, and improved health behaviors (*Roberts, 1999*); and indirectly, through improved access to and use of health services, with evidence of decreased utilization, improved self-education (*Collins, Bybee & Mowbray, 1998*), and improved mental health outcomes (*Taub, Tighe & Burchard, 2001*). Advocacy-based mental health empowerment programs put patients in positions of service, improving their quality of life and social support. They can also lead to changes in practice and policy, such as better recreation services (*Gammonley & Luken, 2001*), new respite facilities, anti-stigma coalitions, and consumer rights policies (*Hess et al., 2001*).

The key components of the current PE strategies for patients with CVD include shared decision making, patient involvement, health literacy, and self-management.

### Self-management

Social services and healthcare are essential for helping individuals with long-term illnesses. They offer equipment, drugs, and expert personnel to address symptoms. Beyond receiving medical care and counseling, people with illnesses like CVD must manage their condition in their daily lives.

The ability of a person to control symptoms, medication, treatment regimens, side effects (both physical and psychological), and lifestyle modifications that come with having a chronic illness is known as self-management (*Sol et al., 2006*). Regretfully, a lot of patients lack the self-assurance necessary to manage their CVD properly (*Satink et al., 2013*). Programs for self-management can aid in closing this gap. These initiatives help patients recognize problems, remove obstacles, get help, develop solutions, and set and keep track of short- and long-term objectives (*McGowan, 2005*). Patients can regain control over their health by actively engaging in their care.

Positive effects can result from successful self-management programs that foster self-efficacy and behavioral change. The most successful integration of these programs into larger projects includes information sharing, technology-enabled symptom monitoring, online peer support, and behavioral and psychological therapies (*Lorig & Holman, 2003*). Face-to-face self-management programs have demonstrated efficacy in the past, but they mainly depend on the patient's ability to control their health (*Sol et al., 2005*, *2011*; *Janssen et al., 2013*). Web-based programs anticipate filling this gap. According to published research, these initiatives may have a positive impact on clinical outcomes, decrease risk factors, improve secondary prevention, decrease hospital admissions, and lower mortality rates (*Ekeland, Bowes & Flottorp, 2010*; *Clark et al., 2007*; *Neubeck et al., 2009*; *Holland et al., 2005*).

CVD self-management education is defined as a collaborative, continuing process meant to assist the development of knowledge, skills, and abilities necessary for successful self-management of CVD by the critical criteria for CVD self-management education and support. It is imperative to motivate individuals with chronic diseases to accomplish effective self-management, including lifestyle modification (*Lambrinou, Hansen &*

*Beulens, 2019*). The three leading causes of non-adherence to efficient self-management are exhaustion associated with the disease, lack of social support, and lack of drive. Providing patients with CVD with self-management education and resources is essential. Individual needs should be taken into consideration when designing self-management education programs. These include the individual's medical history, age, health beliefs and attitudes, knowledge, health literacy, physical limitations, support from family and friends, financial situation, and sociocultural factors (*Bagnasco et al., 2013*; *Gopalan et al., 2018*). Individuals with CVD should consider their living environment, including their physical and social surroundings, and their traits (*Schmuhl et al., 2014*).

### Health literacy

Comprehending health information is also essential to patient empowerment. According to *den Broucke (2014)*, it is defined as people's ability to access, comprehend, evaluate, and apply health information to make decisions about healthcare, illness prevention, and health promotion in daily life to maintain or enhance quality of life.

In a recent scientific statement, the American Heart Association emphasized the vital role that health literacy plays in cardiovascular disease (*Magnani et al., 2018*). Access to healthcare resources, relationships between patients and providers, and self-care behaviors are all significantly impacted by health literacy (*Paasche-Orlow & Wolf, 2007*). Acquiring knowledge and skills related to self-care can help mitigate the adverse health effects associated with low health literacy levels, given its strong connection to social determinants of health, such as education, income, language barriers, and other physical, cultural, and environmental factors.

Digital methods can benefit from health literacy, but health technology has yet to live up to its potential to improve health outcomes (*Burke et al., 2015*).

Digital methods provide the flexibility to evaluate understanding, eliminate time limits, and clarify or expand on complex subjects (*Conard, 2019*). Showing how health technology may improve health literacy, track health measures, and lower healthcare costs is one strategy that shows promise. Health literacy is a proxy marker, but it has a history of accurately indicating improvements in self-care practices, involvement with healthcare, and health outcomes. Prompting health technology providers to demonstrate the efficacy of their offerings can stimulate significant funding toward the continuous investigation and advancement of more effective health technology methodologies. Creating experiences that are action-oriented, personalized, relevant, and interactive is emphasized in the best practices for digital health literacy. Blockchain, machine learning, virtual and augmented reality, artificial intelligence, and other emerging technologies have the potential to advance technology beyond simple data collection and create a more cohesive system. Digital solutions enable individuals to actively participate in their health rather than just being passive consumers of healthcare information. Persons now have more control over their health, better access to their health information, and enhanced communication with their healthcare team due to this move towards a more person-centered approach.

In comparison to traditional doctor-patient encounters, digital tools, wearables, and smart devices offer additional benefits, such as the ability to deliver multimedia education,

collect biometric and subjective data related to medication and symptom management, monitor nutrition and other behaviors, and facilitate communication outside of the healthcare setting (*Conard, 2019*).

A complicated phenomenon known as "patient involvement" occurs when people create meanings and behaviors representing the level of involvement they would like in individual and social decision making (*Tritter & McCallum, 2006*). According to this viewpoint, patient participation encompasses decision making. It is characterized by possibilities for patients to engage in various healthcare activities at varying degrees of involvement across various organizational levels. Patient engagement is a dialogical, interactive, and cooperative process grounded in the knowledge and experience of patients, their relatives, and healthcare providers regarding actions that impact patients' health (*Gustavsson, 2016*). It also has a lot of advantages, such as empowering patients to take a more active role in their health management by using technology to communicate with other patients and share knowledge. In terms of creating and putting into practice patient empowerment, it also delivers notable productivity gains. People desire more significant involvement in issues about their care and health (*Q Care Quality Commission, 2016*).

Crucially, patient participation in CVD management should be viewed as an integral component of healthcare rather than a stand-alone activity. Healthcare personnel, the more extensive health system, and patient education and preparation are all important in the context of patient empowerment for CVD. Encouraging patients to take an active role in their care requires educating them about their condition, available treatments, lifestyle changes, and self-management strategies. The nature of CVD, risk factors, medication treatment, symptom detection, and the significance of lifestyle modifications should all be included in this education. Educative materials that are easily understandable and accessible, such as pamphlets, films, and Internet sources, can improve patients' comprehension and involvement. Healthcare practitioners' understanding of CVD management, patient-centered communication, shared decision making, and motivational interviewing approaches should be improved through training and professional development programs. Healthcare personnel can more successfully address patients' problems, involve them in decision making, and communicate with them when prepared. To manage CVD effectively, patient empowerment requires collaboration between patients, healthcare professionals, and support systems. Treatment plans can better reflect patients' preferences, values, and objectives when patients and healthcare providers participate in decision making. Encouraging patients to participate in conversations, pose inquiries, and offer comments cultivates a feeling of joint responsibility and collaboration in their medical journey.

### Shared decision making

Individuals are supported to participate in the decision making process to the extent they choose through shared decision making. With shared decision making, individuals receive assistance in comprehending the range of available care, treatment, and support options, as well as each option's advantages, disadvantages, and implications. They then select a preferred course of action based on their preferences and high-quality, evidence-based

information (*Team within the Personalised Care Group at NHS England, S. D. M, 2019*). Thus, it is a collaborative process wherein patients and doctors choose therapies, tests, care plans, or other resources based on available data and the patient's well-informed preferences (*Coulter & Collins, 2011*). By ensuring that people are encouraged to make decisions according to their preferences, shared decision making increases the likelihood that people will follow evidence-based treatment plans, have better results, and feel less regret after making their choices (*Stacey et al., 2014*). Evidence-based decision support tools designed to help people (particularly those with low health literacy) understand their options and what is known about the benefits, harms, consequences, and burdens of those options can be used to support the process (*Team within the Personalised Care Group at NHS England, S. D. M, 2019*).

Table 1 provides a summarized view of the four primary aspects of patient empowerment strategies, their goals, and the common challenges associated with each strategy that are described above.

## Impact of 5P-Medicine on PE strategies

The following sections provide a detailed yet straightforward explanation of the charts that illustrate the impact of the 5P-Medicine approach on patient empowerment. These visuals and their interpretations enhance the understanding of how patient empowerment improves through the 5P-Medicine strategies.

The data used for creating the charts was extracted from the literature review on the effectiveness of 5P-Medicine strategies on patient empowerment. The strategies included in the analysis were Predictive, Preventive, Participatory, Personalized, and Precision. The empowerment scores before and after the implementation of 5P-Medicine were compiled from the various studies included in the review. The scores were standardized and converted into percentages to facilitate comparison.

The collected data was analyzed to determine the mean empowerment scores for each strategy before and after the implementation of 5P-Medicine. It involved calculating the average scores reported across different studies to provide a single representative value for each strategy.

The bar chart presented in Fig. 2 compares patient knowledge levels across four key strategies, such as self-management, health literacy, patient involvement, and shared decision making, before and after implementing the 5P-Medicine approach.

Self-management increased knowledge from 35% to 70%, indicating that patients became more informed about managing their health independently. Health literacy rose from 45% to 85%, significantly improving patients' understanding of health information and resources. Patient involvement jumped the patient's knowledge from 40% to 80%, suggesting that patients became more aware of their roles in healthcare decisions. Shared decision making improved it from 30% to 65%, highlighting that patients are better informed about their options and can actively participate in treatment choices.

The 5P-Medicine approach effectively enhances patient knowledge across all strategies, empowering patients to make informed decisions about their health.

**Table 1  Main strategies of patient empowerment.**

| Strategies | Techniques | Goals | Challenges |
|---|---|---|---|
| Self-management | Programs facilitating self-care, goal-setting, and skill development. | Empower patients for active disease management, lifestyle changes, and adherence to treatment plans. | Lack of patient confidence, motivation, social support, and disease-related barriers. |
| Health literacy | Education and access to health information and resources for informed decision making. | Enable patients to comprehend health information, engage in healthcare decisions, and improve outcomes. | Socioeconomic disparities, language barriers, and limitations in understanding health information. |
| Patient involvement | Collaboration, shared decision making, and engagement in healthcare activities. | Encourage patients to actively participate in their care, fostering trust, satisfaction, and adherence. | Resistance from healthcare providers, time constraints, varying health literacy, and cultural issues. |
| Shared decision making | Jointly deciding on treatment plans based on patient preferences and evidence-based information. | Support patients in making decisions aligned with personal preferences, increasing treatment adherence. | Implementing evidence-based decision support tools, diverse health literacy levels, and time constraints. |

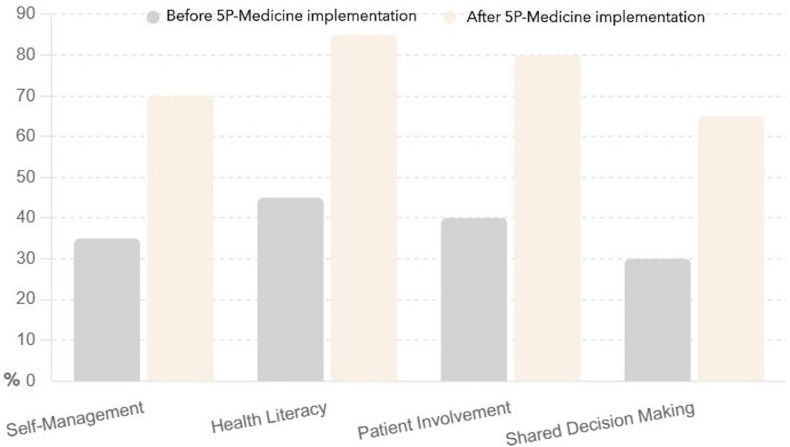

**Figure 2  Patient knowledge before and after 5P-Medicine implementation.**

This bar chart presented in Fig. 3 illustrates the improvement in patient skills across the same four strategies before and after adopting the 5P-Medicine approach.

Self-management increased patients' skills from 30% to 68%, indicating that patients developed better abilities to manage their health conditions.

Health literacy improved patients' self-management and engagement from 40% to 83%, showing a significant improvement in patient's ability to understand and use health information. Patient involvement improved from 35% to 78%, suggesting that patients became more capable of actively participating in healthcare processes. Shared decision making advanced these from 25% to 65%, highlighting that patients are more equipped to engage in discussions and decisions about their treatment options.

The 5P-Medicine approach significantly improves patient skills, enabling them to manage their health and engage in their healthcare effectively.

This bar chart presented in Fig. 4 highlights the increase in patient confidence across the four strategies before and after the implementation of the 5P-Medicine approach.
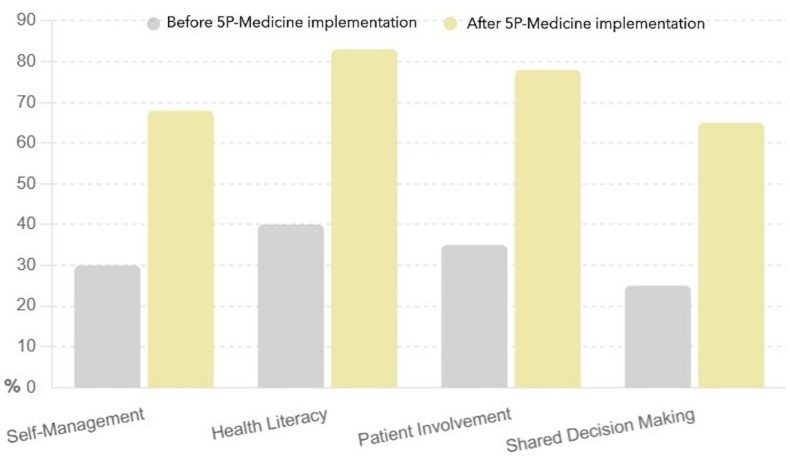

**Figure 3** **Patient self-management and engagement before and after 5P-Medicine implementation.**

The confidence in self-management increased from 25% to 65%, indicating that patients feel more assured in managing their health independently.

Next, the confidence of health literacy rose from 35% to 80%, showing a significant improvement in patients' confidence in understanding and using health information.

The confidence of patient involvement improved from 30% to 75%, suggesting that patients feel more capable of participating in healthcare decisions.

Finally, the confidence in shared decision making increased from 20% to 60%, highlighting that patients are more confident in discussing and deciding on their treatment options.

The 5P-Medicine approach greatly enhances patient confidence, empowering them to take charge of their healthcare more assuredly.

Implementing the 5P-Medicine approach, which includes Predictive, Preventive, Participatory, Personalized, and Precision aspects, profoundly impacts patient empowerment. Each chart shows substantial improvements in knowledge, skills, and confidence across four key empowerment strategies: self-management, health literacy, patient involvement, and shared decision making.

This bar chart presented in Fig. 5 illustrates a clear and substantial improvement in patient empowerment across all five strategies after the implementation of 5P-Medicine. The most significant increases were observed in the Personalized and Participatory strategies, where empowerment scores rose by 25% and 35%, respectively. It suggests that personalized care and active patient participation are particularly effective in enhancing patient empowerment. The Preventive strategy also saw a notable increase of 20%, indicating that proactive health measures contribute significantly to patient empowerment. The Predictive and Precision strategies both saw substantial improvements of 20% and 28%, respectively, highlighting the importance of predictive analytics and precision medicine in empowering patients.

Comprehensive, the 5P-Medicine strategies have a profound positive impact on patient empowerment, leading to higher engagement, better self-management, and more active
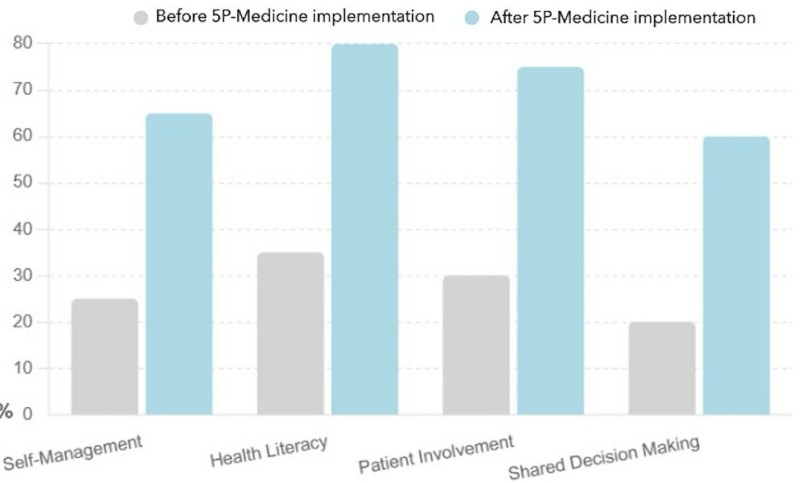

**Figure 4 Patient confidence before and after 5P-Medicine implementation.**

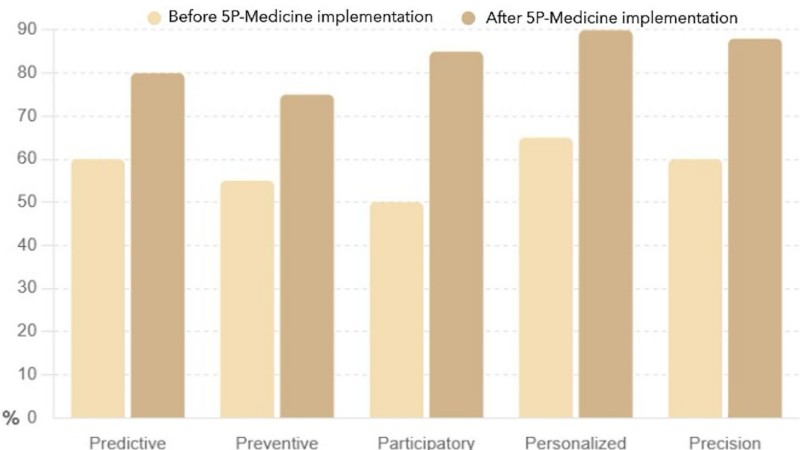

**Figure 5 Patient empowerment before and after 5P-Medicine implementation.**

involvement in healthcare decisions. This visual and narrative description underscores the effectiveness of the 5P-Medicine approach in improving patient outcomes.

## Comparison of PE with other health conditions

In terms of patient empowerment, patients with CVD may differ from patients with other medical diseases, such as diabetes, cancer, and stress-related disorders, in a few specific ways. These traits may influence how patient empowerment techniques are implemented.

As per *Libby & Theroux (2005)*, CVD generally affects the heart and blood vessels, while cancerous illnesses mainly entail abnormal cell growth in different organs or tissues (*National Care Institute, 2008*). While stress-related disorders affect the nervous system and the body's reaction to stress (*American Institute of Stress, 2023*), diabetes primarily affects the endocrine system, specifically insulin regulation and blood sugar control (*World Health Organization, 2022b*). Different kinds of tests are used in the diagnosis procedure

for different illnesses. Tests like electrocardiograms (ECG/EKG), stress tests, echocardiograms, and angiographies are used in CVD diagnosis procedures. Imaging methods such as X-rays, biopsies, magnetic resonance imaging (MRI), and computed tomography (CT) scans are frequently needed for oncological disorders (*Jafari et al., 2020*). Blood tests that measure blood sugar levels are frequently used to diagnose diabetes (*Khan et al., 2019*). Psychological evaluations, assessments of stressors, and the exclusion of other possible medical explanations are all possible steps in treating stress-related illnesses (*Bremner et al., 2020*).

Therapies for CVD include lifestyle changes (diet, exercise, stopping smoking), drugs to control risk factors (high blood pressure, high cholesterol), cardiac procedures (bypass surgery, angioplasty), and cardiac rehabilitation (*Arnett et al., 2019*). To manage underlying illnesses such as diabetes and obesity, preventive approaches address risk factors and encourage healthy lifestyles (*Germano et al., 2012*). Treatments for oncological disorders vary greatly and are primarily based on the kind, stage, and location of the cancer, as well as the general health status of the patient. These could consist of immunotherapy, hormone treatment, targeted therapy, radiation therapy, chemotherapy, or a mix of them (*Wang, Lei & Han, 2018*). Regarding prevention, early detection and lifestyle modifications are frequently at the forefront. It may include routine screenings, self-examinations for specific malignancies, abstaining from tobacco and UV radiation, maintaining a healthy weight and diet, exercising regularly, and consuming less alcohol (*Schiffman, Fisher & Gibbs, 2015*).

Additionally, vaccinations are essential in avoiding certain cancers. For example, the Hepatitis B vaccine prevents liver cancer, and the Human Papillomavirus (HPV) vaccine prevents cervical and other HPV-related cancers (*Finn, 2003*). Treatment strategies for diabetes usually include regular blood sugar testing, insulin therapy, oral drugs, and lifestyle changes (diet, exercise, weight management). Changes in lifestyle, regular check-ups, and screening for prediabetes or early identification of diabetes complications are examples of preventive approaches (*American Diabetes Association, 2020*). Treatments for stress-related disorders involve lifestyle modifications (such as exercise, a healthy lifestyle, good sleep hygiene, and mindfulness), stress management strategies (such as relaxation exercises and mindfulness), and, in certain situations, medication for symptomatic relief. Prevention tactics aim to lower stress, increase resilience, and encourage constructive coping techniques. They include regular self-care routines like physical activity, enough sleep, a balanced diet, and social interaction. It can also be advantageous to develop constructive coping strategies, such as assertiveness, time management, and problem-solving techniques (*World Health Organization, 2022a*).

Each health condition has its unique characteristics, diagnostic processes, treatment approaches, preventive measures, and challenges, influencing the specific strategies for empowering patients in managing their health. Table 2 provides a comparative overview of the main aspects of CVD in contrast to other health conditions like oncological conditions, diabetes, and stress-related conditions.

Each patient's needs and specific health conditions influence patient empowerment initiatives. Cardiovascular illness emphasizes a lifetime commitment to lifestyle

**Table 2 Overview of patient empowerment strategies in the context of cardiovascular disease compared to oncological conditions, diabetes, and stress-related conditions.**

| Aspect | Cardiovascular disease (CVD) | Oncological conditions | Diabetes | Stress-related conditions |
|---|---|---|---|---|
| Disease characteristics | Affects the cardiovascular system, including the heart and blood vessels | Involves abnormal cell growth in various organs or tissues | Predominantly affects the endocrine system, insulin regulation | Impacts the nervous system and body's response to stress |
| Diagnosis procedure | ECG/EKG, stress tests, echocardiogram, angiography | Imaging (X-rays, CT scans, MRI), biopsies | Blood tests measuring blood sugar levels | Psychological assessments, stress trigger evaluation |
| Treatment approaches | Lifestyle modifications, medications, cardiac procedures, rehabilitation | surgery, radiation therapy, chemotherapy, targeted therapy, immunotherapy, hormone therapy | lifestyle modifications, medications, insulin therapy | Stress management techniques, lifestyle changes, medication |
| Preventive measures | Focus on addressing risk factors, promoting heart-healthy lifestyles, managing underlying conditions | Early detection, vaccines, screenings, self-examinations | Lifestyle changes, routine check-ups, screening for prediabetes | Stress reduction, strengthening resilience, healthy coping mechanisms |
| Patient empowerment strategies | Emphasizes lifelong commitment to lifestyle changes, education about self-monitoring, and medication management | Requires comprehensive education about the condition, treatment options, coping strategies, and significant emotional support | Involves daily self-care exercises, including monitoring blood glucose, regulating diet, and using technology | Focuses on inculcating effective coping mechanisms, stress management techniques, and cognitive-behavioral interventions |
| Unique challenges and considerations | Long-term commitment to lifestyle changes, addressing co-morbidities, and multifaceted approach due to a large and growing patient population | Complex treatment regimens, the need for comprehensive education, and significant emotional support | Daily self-care exercise, technology literacy, and a focus on long-term management | Requirement for effective coping mechanisms, stress management, and attention to mental health aspects |

modifications, with the most critical component being education regarding self-monitoring and medicine. Patients with cancer who must adhere to complicated treatment plans require a great deal of emotional support in addition to thorough information about their disease, available treatments, and coping mechanisms. Diabetes management is a daily self-care practice involving technology, controlling nutrition, and checking blood sugar. Patient literacy of their condition and course of treatment is necessary. Empowerment tactics that impart efficient coping mechanisms, stress management approaches, and cognitive-behavioral therapies are necessary for treating disorders related to stress. All illnesses require an informed and proactive patient to collaborate with healthcare experts to obtain the best possible health results.

The disorders associated with stress, diabetes, cardiovascular disease, and cancerous illnesses are significant contributors to the worldwide burden of disease. These are common illnesses that cause significant morbidity and mortality, impacting a sizable section of the populace. Choosing these particular medical conditions offers a thorough understanding of patient empowerment, considering the various physiological systems, diagnostic procedures, treatment modalities, preventive measures, and the long-term commitment necessary for successful management.

This study highlights that comparing these circumstances does not provide a one-size-fits-all approach to patient empowerment. Customized strategies that consider each health

condition's distinct qualities, difficulties, and requirements are needed to obtain the best possible health results.

To empower CVD patients, the demands of a sizable and expanding patient community must be met. This calls for a multidimensional strategy that considers this patient population's particular needs and difficulties. It offers resources, knowledge, and tools to assist patients in effectively managing their conditions, addressing co-morbidities, and enhancing their social, mental, emotional, and physical well-being.

## EMERGING TECHNOLOGIES USING FOR PATIENT EMPOWERMENT

Cardiovascular disorders are often disregarded, but they might have unexpected, sometimes fatal effects for which emergency care is needed (*Singh et al., 2014*). Early on, they frequently come with notable alterations in vital signs. Thus, it is critical to enhance daily health monitoring to prevent and treat heart disease. Patients require adequate technology that allows them to authorize the use of their data to increase their access to and control over their health. Technologies like artificial intelligence, machine learning, wearable and smart device connectivity, sensor design, data capture, and the Internet of Things (IoT) allow patients to monitor their vital signs and healthcare issues anytime and anywhere. Wearable health technology is a patient and consumer phenomenon as powerful as the transition in medical practice toward patient-centered care.

### Information and communication technologies

The introduction of new technology is impacting the delivery of healthcare. The creation of innovative healthcare models that will benefit patients and the medical community is being aided by new medical technologies. Quick and dependable ICTs are now necessary for raising the standard of healthcare globally since they offer a crucial chance to enhance coordination throughout the whole healthcare delivery system (*Kierkegaard, 2012*).

Information and communication technologies (ICT) are essential to patient empowerment by simplifying information interchange, granting patients access to healthcare resources, and encouraging active patient participation in their care.

ICT for health has resulted in various applications across programs, including e-prescriptions, patient portals with electronic health records (EHRs), and mobile applications. They have brought about a significant transformation in the delivery of healthcare. Patients now have access to many healthcare data, protecting patient privacy and security and enhancing the precision and safety of healthcare delivery. Patients are empowered by mobile apps created specifically for the healthcare industry, which give them access to various tools and services. Medication reminders, symptom logs, health journals, activity tracking, and access to instructional resources are just a few of the functions that these applications may offer. Through mobile apps, patients may take control of their health, monitor their development, and get individualized guidance and assistance. Secure websites or programs that give patients access to their test results, medical information, and appointment scheduling are known as online or patient portals (*Lemire, Sicotte & Paré, 2008*). Patients can request prescription refills, monitor their

medical history, access their health information, connect with healthcare professionals, and participate in joint decision making through these portals. By giving patients electronic reminders and notifications, e-prescription systems can enhance drug adherence (*Shaw & Baker, 2004*). To help patients keep on track with their medications and lower the chance of missing doses, patients can receive reminders about prescription refills, drug regimens, or dosage instructions. EHRs and pharmacy databases are just two examples of the different healthcare systems that the E-rescription systems can interface with. The smooth data exchange made possible by this integration lowers the possibility of drug errors or duplicate prescriptions while improving care coordination.

With telemedicine and remote monitoring technologies, patients can check their health conditions and receive virtual consultations from the comfort of their homes, removing obstacles like geographic location, transportation problems, or a shortage of healthcare providers. Information technology facilitates contact between medical professionals at one site and patients or providers at another, known as telemedicine (*Craig & Petterson, 2005*). Improving communication fosters a collaborative relationship, and patients can feel heard, respected, and actively involved in their care. Better results result from patients more likely to follow their treatment programs. For patients with chronic diseases, remote monitoring makes monitoring and follow-up care easier (*Alvarez et al., 2021*). Patients can use wearable technology or home monitoring equipment to gather and send important health data to healthcare providers. It makes it possible to monitor health conditions continuously, identify changes or difficulties early, and take appropriate action when needed. These technologies empower patients through more access to specialized care, convenient monitoring and follow-up, patient education and self-management, improved communication, and lower healthcare costs. Technologies like telemedicine and remote monitoring improve patient outcomes, increase patient participation, and support a patient-centered approach to cardiovascular care.

Patients can obtain accurate and evidence-based information about their health issues, treatment options, and self-care practices by accessing reputable health education websites and online resource platforms, materials, interactive modules, and multimedia resources (*Lemire, Sicotte & Paré, 2008*). Patients can actively participate in shared decision making with healthcare practitioners, educate themselves, and better understand their health.

Online health communities and support groups allow people to interact, exchange stories, and seek help from others with comparable medical issues (*Eysenbach et al., 2004*). These communities empower patients by offering a forum for information sharing, peer education, and emotional support, which lessens isolation and promotes a sense of belonging.

ICTs empower patients through better access to healthcare services, health information, and resources, improved health literacy, patient-provider communication, improved self-management, and peer support. They allow patients to participate in their well-being, make educated decisions, and participate in their healthcare.

## Wearables and smart devices

Consumer-grade, networked electronic gadgets known as "smart wearables" are integrated into clothing or worn as accessories on the body. These encompass a variety of devices, such as smartwatches, rings, and wristbands, among others. They are equipped with multiple advanced sensors and powerful computing capacity, enabling them to extract novel health insights (*Bayoumy et al., 2021*). Real-time monitoring of human physical activities and behaviors and physiological and biochemical data are made possible by wearables and smart devices (*Prieto-Avalos et al., 2022*). Electrocardiogram (ECG), electromyogram (EMG), heart rate (HR), body temperature, electrodermal activity (EDA), arterial oxygen saturation (SpO2), blood pressure (BP), blood glucose (BG), blood cholesterol level (BCL), and respiration rate (RR) are among the physiological signs that wearables can measure through embedded sensors (*Majumder, Mondal & Deen, 2017*; *Reda et al., 2021*).

The medical profession acknowledges numerous wearables and smart devices as valuable resources for illness diagnosis, monitoring, and treatment. The most widely used ones are fitness trackers, smartwatches, smart blood pressure monitors, smart contact lenses, pulse oximeters, sleep trackers, ECG monitors, smart inhalers, and smart pillboxes. There are numerous examples on the market, including:

- Apple Watch, Fitbit, and Garmin devices have features like stress management, activity tracking, heart rate monitoring, and sleep analysis;
- Dexcom G6 and Freestyle Libre are frequently used by individuals with diabetes to continuously monitor their blood sugar levels and provide real-time data, which can aid in effectively managing blood sugar levels;
- Users can measure and monitor their blood pressure at home with the Withings BPM Core and Omron Evolv devices.

Those with hypertension or those keeping an eye on their cardiovascular health can benefit from them; iv) The gadgets, such as those produced by Nonin or Masimo, measure the blood's oxygen saturation levels. In addition to being useful during exercise or at high altitudes, they are frequently used by people with respiratory conditions like asthma or chronic obstructive pulmonary disease (COPD); v) Devices like Oura Ring and Beddit use sensors to monitor sleep patterns, including duration, quality, and stages of sleep. vi) Users can record electrocardiogram readings with portable ECG monitors, such as AliveCor's KardiaMobile or Apple Watch's ECG function. These devices can offer insights into sleep abnormalities and aid in identifying probable sleep disorders. These are useful for people who have heart problems or are at risk of having a cardiac attack; vii) Teva's ProAir Digihaler or Propeller Health's sensors are connected to inhalers and offer advice on how to use the device and how much medication to take at a time. They can aid in the better management of asthma and chronic obstructive pulmonary disease (COPD); viii) Pillsy and MedMinder are two examples of devices that use sensors and reminders to assist users in adhering to their prescription regimens. They can lower the possibility of missing doses by sending out alerts and reminders to take prescribed drugs on time.

These wearables and smart gadgets aim to improve self-care and empower patients. They may offer insightful information and valuable data to help preventative health care. Medical specialists should always use them to guarantee proper medical advice and decision making. It is crucial to remember that the medical community's acceptance of these gadgets may differ depending on particular use cases and the inclinations of specific healthcare organizations or providers.

Furthermore, advancements in wireless sensor networks (WSN), artificial intelligence (AI), and the Internet of Things (IoT) have emerged as crucial allies in the creation of mHealth wearable devices. AI and neural networks are being incorporated into wearables using deep learning and signal processing techniques (*Prieto-Avalos et al., 2022*) to increase the accuracy of ECG and PPG signals and the dependability of ambulatory monitoring devices. Wearable technology performs better, and biological variable assessments for CVD are more accurate thanks to artificial intelligence and neural networks. Therefore, consumer adoption of wearable devices rises with their increasing accessibility and dependability (*Nahavandi et al., 2022*). Wearable-based technologies are becoming a more practical option for continuously monitoring CVD patients' health while they go about their daily lives.

## Artificial intelligence

AI applications in health innovation are increasing, indicating how these technologies might be used to solve issues like worker shortages and aging populations. It can lower healthcare expenditures while improving patient care significantly. Rapid advancements in artificial intelligence and robotics in healthcare are being made, particularly in early detection and diagnostic applications (*Coeckelbergh, 2010*). AI has a massive potential for use in the clinical setting, with applications ranging from clinical research and therapeutic decision making to the automation of diagnostic procedures.

The recent exponential growth in artificial intelligence has been fueled by large-scale annotated clinical data collecting, improvements in machine learning techniques, open-source machine learning packages, and inexpensive and quickly expanding computer power and cloud storage.

Medical diagnosis decision making systems and healthcare monitoring applications have long used computational machine learning techniques, including reinforcement, unsupervised, semi-supervised, supervised, and ensemble learning algorithms. Deep learning paradigms have been widely used in biomedical research projects and engineering applications, driven by the burgeoning cloud computing technologies and hardware infrastructures (*Chen et al., 2022*). Based on the encoding and decoding structures, state-of-the-art deep learning neural networks can map spatial data at different levels in numerous network layers and generate localization details of the region of interest. Convolutional neural networks, fully convolutional networks, generative adversarial networks, and recurrent neural networks are the most widely used deep learning architectures (*Wu & Ghoraani, 2022*).

It promises to transform the medical practice environment in a short while. AI systems can estimate patient prognosis more accurately than clinicians, perform several diagnostic

tasks at the specialist level, and support surgical operations. There is a growing belief that AI has the potential to transform medicine and change the responsibilities of clinicians as machine-learning models develop.

## Decision support systems

Information technology systems that provide pertinent data, information, or expertise to help people or organizations make decisions are called decision support systems (DSS). It churns large datasets to give an efficient conduit for decision making. To better organize clinical knowledge and facilitate decision making, most healthcare information systems must either incorporate clinical decision-support practices or currently do so (*Hak, Guimarães & Santos, 2022*). Patients with cardiovascular disease (CVD) may benefit from a DSS in several ways. Comprehensive, intelligible, and pertinent information regarding CVD can be found in a DSS. Information about the illness, its dangers, preventative measures, and available treatments may be included. By increasing their health literacy, patients can better understand their health and the effects of different decisions.

Based on available data, DSS uses machine learning and data analysis methods to forecast future health risks. It might help avert more medical issues. Tools for monitoring pertinent health metrics, including blood pressure, heart rate, weight, physical activity, and food, can be obtained through a DSS. Patients may be motivated to improve their behavior by seeing how their lifestyle decisions and self-management practices affect their health thanks to this continuous monitoring (*Jovicic, Holroyd-Leduc & Straus, 2006*). A DSS can help with shared decision making by giving patients and healthcare professionals a forum to discuss available treatments. An informed discussion on the appropriate course of action can be facilitated by the system's ability to show an individual's health data combined with evidence-based information about various treatment strategies (*Ruland & Bakken, 2002*). A DSS offers individualized guidance by examining a person's health data. For example, depending on a person's particular risk factors and health status, it can recommend specific lifestyle changes, medication adjustments, or more examinations. Some DSSs might offer a peer support network that links patients with other people dealing with similar conditions. It can offer guidance and emotional support, which increases empowerment even more.

A DSS must be carefully designed for CVD management to guarantee that it is dependable, safe, and easy to use. Healthcare professionals should assist patients in using these devices and analyzing their available data. DSS can greatly empower patients with CVD, enabling them to actively participate in their health management, enhance their quality of life, and possibly lower the risk of significant problems.

## Signal processing and data analysis

Signal processing and data analysis techniques are widely used in the field to extract helpful information from the many forms of healthcare data. These methods enhance general patient care, monitoring, diagnosis, and treatment in several areas related to patient empowerment. A few of these frequently used methods are examined here.

Physiological signals are frequently analyzed using signal-processing techniques in various domains, including biotechnology, neuroscience, and Medicine. Electrical or biological signals produced by the human body or other living things are called physiological signals, and they provide important information on the health and functions of the body. Electrocardiograms (ECG), electroencephalograms (EEG), and respiratory signals are among the health-related signals from which noise and artifacts are removed using signal processing techniques, including filtering (*Colditz, Burke & Celka, 2001*).

Relevant features are also extracted from health signals using signal processing techniques. In the context of ECG analysis, techniques such as wavelet transforms, heart rate variability (HRV) analysis, and QRS complex recognition, for instance, extract features that shed light on cardiac health and problems.

Signal compression techniques decrease the storage and transmission space needed for health-related signals and images. Utilizing this growing amount of data and exploring and presenting the information appropriately for medical purposes requires medical image processing.

Medical imaging modalities include computed tomography (CT) scans, ultrasound, magnetic resonance imaging (MRI), and digital pathology, frequently using signal processing techniques (*Ritter et al., 2011*) to extract pertinent characteristics, segment regions of interest, and improve picture quality. Image processing techniques such as edge detection, segmentation, registration, and filtering facilitate medical picture visualization, interpretation, and analysis. Health data may be efficiently stored, transmitted, and analyzed thanks to compression techniques, eliminating unnecessary or duplicate information.

Health signals are subjected to spectral analysis techniques like wavelet and Fourier transforms to determine their frequency content. By identifying particular frequency components, patterns, and signal anomalies, spectral analysis facilitates diagnosing and monitoring neurological disorders, cardiovascular diseases, and sleep problems.

Healthcare data can be analyzed using various strategies and tactics to glean insights. These methodologies employ statistical analysis, machine learning, and further computational approaches to examine many forms of healthcare data, including patient-generated data, medical imaging data, electronic health records (EHRs), and genetic data.

Healthcare organizations frequently employ machine learning (ML) techniques to evaluate and interpret large, complicated datasets, which eventually help patients take charge of their care. To create individualized treatment regimens, machine learning algorithms can evaluate patient-specific data, such as clinical factors, genetic information, and electronic health records. Machine learning models let medical professionals customize treatments for each patient by finding patterns, correlations, and treatment responses in massive datasets. Precision medicine is a method that empowers patients with therapies that are more effective and have fewer side effects, allowing for patient-centric care. Machine learning algorithms examine data from wearable devices in various healthcare applications to track and understand patients' health states. Machine learning algorithms can identify abnormalities, forecast health concerns, and give patients

immediate feedback by continuously evaluating physiological data, activity levels, and sleep patterns. It empowers people to take proactive steps to avert unfavorable health events, make educated lifestyle decisions, and actively monitor their health. Supervised learning algorithms are used for illness prediction, risk stratification, and treatment outcome prediction, including decision trees, random forests, support vector machines, and neural networks (*Alanazi, 2022*). Based on unsupervised learning methods, including dimensionality reduction and clustering, it aids in finding patterns and assembling related individuals or illnesses.

Information from unstructured healthcare data, such as clinical notes, medical literature, and patient questionnaires, is extracted and analyzed using natural language processing (NLP) techniques to improve patient empowerment. NLP algorithms can extract pertinent clinical data, recognize important terms or concepts, and aid in the semantic interpretation of textual data (*Friedman & Elhadad, 2014*). NLP makes it possible to use free-text data for text mining, sentiment analysis, information retrieval, and clinical coding. Using natural language processing (NLP) tools to extract and evaluate information from unstructured healthcare data can enhance patient empowerment in several ways. Personalized insights from clinical data can help patients and healthcare providers communicate better and make decisions together. Patients with access to current and pertinent medical material are better equipped to participate actively in discussions about their treatment options. Furthermore, by better understanding and addressing patient issues, patient feedback analysis can assist healthcare companies in raising patient happiness and engagement levels.

Data mining techniques can examine and identify links and patterns in big datasets. Classification, clustering, and association rule mining algorithms are used to extract hidden insights from healthcare data. Data mining techniques have significantly enhanced the medical industry's capacity for prediction and decision making about diseases such as diabetes, heart disease, liver disease, cancer, and others (*Patel & Patel, 2016*). Data mining facilitates the identification of variables affecting patient outcomes, the discovery of co-occurrence patterns, and the generation of research hypotheses.

Techniques for time series analysis are used on data gathered over an extended period, such as patient vital signs, the course of a disease, or healthcare utilization. Time series forecasting models have been successfully used in medical applications to estimate the mortality rate, predict the course of the disease, and evaluate time-dependent risk (*Bui et al., 2017*). Time series techniques allow for identifying patterns, seasonality, and future value predictions (*Ewusie et al., 2020*). These techniques include exponential smoothing, Fourier analysis, and autoregressive integrated moving averages (ARIMA). These methods can benefit the healthcare industry by monitoring patient situations, anticipating disease outbreaks, allocating resources wisely, and promoting patient empowerment.

## Data security and privacy protection solutions

Because medical data is susceptible and private, data security and privacy protection solutions are essential to healthcare technologies and patient empowerment.

Patient data privacy in healthcare is a moral, legal, and ethical requirement. Patient data protection is required by laws like the General Data Protection Regulation (GDPR) in Europe and the Health Insurance Portability and Accountability Act (HIPAA) in the United States. Adhering to these requirements is imperative to prevent legal consequences and uphold ethical principles.

Patients with cardiovascular disease use ICT tools to exchange private medical information with healthcare professionals. Vulnerabilities in the platforms that produce and send data can be used to leak information or even take down devices. Strong patient data encryption can help to lessen it (*Segura Anaya et al., 2018*; *Filkins et al., 2016*). Concerns about user privacy also exist since wearable data can be utilized to obtain crucial user information that could be used for commercial purposes (*Jiang & Shi, 2021*). Research suggests that consumers might need to be aware of the privacy hazards associated with these gadgets, emphasizing the necessity of instructional programs. Additionally, the level of access to personal data consented to during the setup of these devices and the acceptance of the conditions of use was not exact. These issues can be addressed with regulations emphasizing data security, open privacy practices, and the appropriate implementation of the informed consent procedure (*Kumar et al., 2022*).

Patients need to feel comfortable knowing that their sensitive medical information is protected. When they have confidence that their private health information is secure, patients are more inclined to actively participate in their healthcare and share critical information with healthcare practitioners.

## Emerging technologies applications and benefits: overview

Table 3 reflects a summarized view of emerging technologies, their applications, and their benefits regarding patient empowerment.

# CONNECTION OF PATIENT EMPOWERMENT WITH A 5P-MEDICINE

A vital component of the 5P-Medicine approach is patient empowerment. Predictive, preventive, participatory, personalized, and precision medicine are all included in 5P-Medicine (*Blobel, Ruotsalainen & Giacomini, 2022*), each with a distinct value. They strengthen and complement one another when combined. 5P-Medicine is a healthcare approach that considers individual genetic profiles and environmental and lifestyle aspects to give more precise and tailored care.

5P-Medicine's predictive component depends on analyzing vast volumes of health data using advanced analytics and machine learning to spot trends that can be utilized to forecast health risks and outcomes. Giving patients access to their medical records and educating them on the consequences of these forecasts can encourage them to take preventative or corrective action against possible health issues.

Furthermore, 5P-Medicine's preventive component concentrates on recognizing and managing health hazards before they become significant issues. Chronic disease and other health disorders can be avoided by providing patients with the information and tools they need to maintain their health and lead a healthy lifestyle.

**Table 3 Emerging technologies for patient empowerment: applications and benefits.**

| Technology | Key applications | Benefits for patient empowerment |
|---|---|---|
| Information and communication technologies (ICT) | Mobile apps, patient portals, E-prescription, telemedicine, online health resources, and patient communities. | Easy access to healthcare information, improved patient-provider communication, ongoing monitoring and follow-up, and access to peer support. |
| Wearables and smart devices | Smartwatches, fitness trackers, CGMs, blood pressure monitors, smart inhalers, *etc.* | Real-time health monitoring, empowerment through data access, and personalized health insights. |
| Artificial intelligence (AI) | Machine learning, deep learning, neural networks. | Enhanced diagnostics and early detection, personalized treatment plans, and precision medicine. |
| Decision support systems (DSS) | Comprehensive health information, risk prediction, self-tracking, shared decision making, and peer support. | Improved health literacy, personalized health advice, and preventing health complications. |
| Signal processing and data analysis | Data extraction, image processing, NLP, data mining, and time series analysis. | Enhanced data interpretation, personalized insights, and disease progression monitoring. |
| Data security and privacy protection | Data encryption, compliance with regulations, and transparency in privacy policies. | Maintain patient data privacy and build patient trust and confidence. |

One essential element of 5P-Medicine's participatory approach is enabling patients to participate actively in their healthcare. It aims to educate and provide the necessary tools while placing the patient at the center of the healthcare system. As a result, the patient can share accountability for their medical care. Healthcare professionals can ensure that treatments and interventions are customized to each patient's unique requirements and preferences by including them in decision making.

The personalized and precision components of 5P-Medicine seek to deliver more individualized and successful therapies by considering each person's genetic, environmental, medical, and lifestyle factors. Their approaches locate the interventions or treatments that enhance the likelihood of a positive outcome by putting the patient at the center of the medical decision making process.

Patient empowerment is a crucial component of 5P-Medicine since it fosters a more patient-centered, customized approach to treatment that considers each patient's unique requirements and preferences. Patient-centered Medicine highlights the limitations of a disease-centered approach to Medicine, namely, the notion that medical professionals should confine their efforts to diagnosing and treating patients' symptoms. In contrast, patient-centered Medicine emphasizes acknowledging and respecting patients' overall life experiences, personal goals, aspirations, and beliefs.

Figure 6 illustrates how the principles of 5P-Medicine, including Predictive, Preventive, Participatory, Personalized, and Precision aspects, are interconnected, with patient empowerment at the core of this healthcare model. Patient empowerment is fundamental to creating a patient-centered approach that aligns with individual patient's values, preferences, and experiences.

## Practical application of the 5P-Medicine

The application of the 5P-Medicine approach has shown promising results in managing cardiovascular diseases (CVD). Integrating these principles into real-world settings
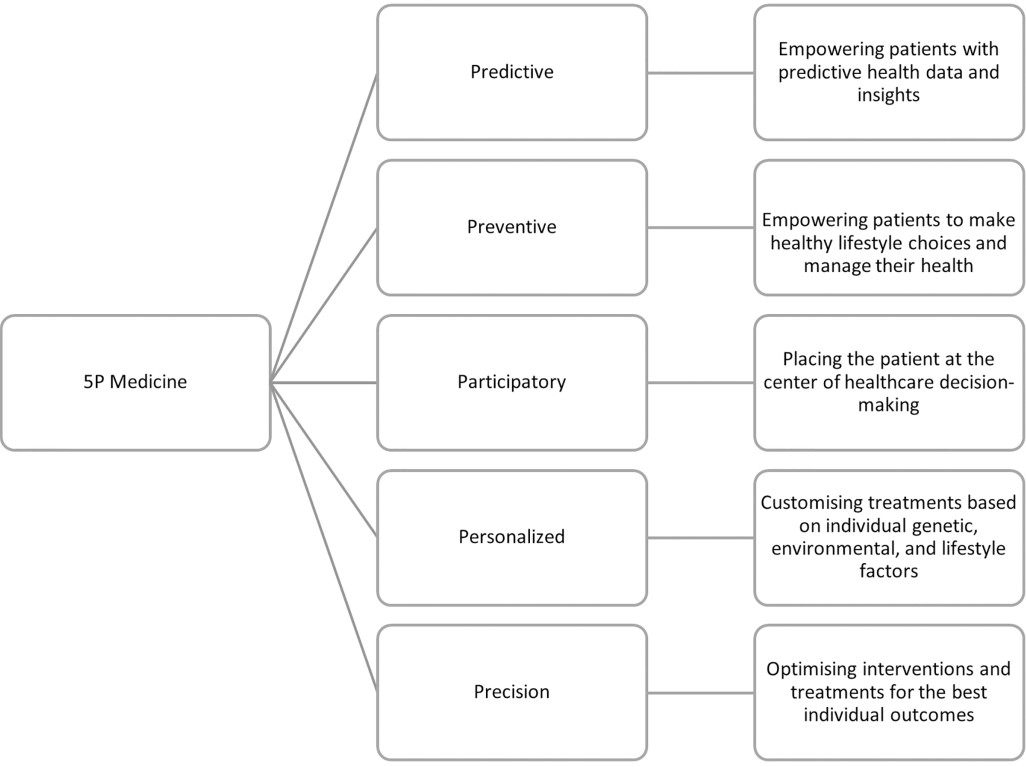

**Figure 6 Patient empowerment is essential to 5P-Medicine, creating patient-centered care.**

demonstrates their effectiveness in improving patient outcomes and empowering individuals to take an active role in their health management. The predictive component leverages advanced analytics and machine learning to identify patterns and forecast health risks. For instance, a study by *Alvarez et al. (2021)* highlighted the use of telemedicine and remote monitoring in heart failure management. Patients using predictive analytics for continuous monitoring experienced fewer hospitalizations and improved management of their condition due to the early detection of potential health issues. Preventive strategies focus on mitigating risk factors before they develop into severe health problems. The authors of *Germano et al. (2012)* reported that lifestyle interventions, such as regular physical activity, dietary modifications, and smoking cessation programs, significantly reduced the incidence of cardiovascular events among high-risk populations. These preventive measures are tailored to the individual risk profiles identified through predictive analytics. Patient involvement in healthcare decisions is a cornerstone of participatory medicine. This study demonstrated that involving patients in their treatment plans through shared decision making processes led to higher satisfaction and better adherence to prescribed therapies. Patients equipped with comprehensive health information were more confident in managing their health, as evidenced by increased engagement in their care plans. Personalized medicine tailors treatments to individual genetic, environmental, and lifestyle factors. The integration of genetic profiling into treatment plans has been shown to enhance the efficacy of interventions. For example, a

clinical trial integrating genetic data to personalize medication for hypertensive patients resulted in significantly better blood pressure control compared to standard treatment protocols. Precision medicine employs targeted therapies to maximize treatment effectiveness while minimizing side effects. Advanced diagnostic tools and precise therapeutic interventions have improved outcomes for patients with complex cardiovascular conditions. A notable example is the use of targeted drug delivery systems that release medication directly to the affected areas, thereby reducing systemic side effects and enhancing therapeutic efficacy.

Implementing the 5P-Medicine approach has profoundly impacted patient empowerment. Studies have shown that patients involved in self-management programs, supported by digital tools and personalized health information, exhibit higher confidence and capability in managing their conditions. Figures 2–4 illustrating patient knowledge, confidence and engagement before and after the implementation of 5P-Medicine show substantial improvements across all empowerment strategies, highlighting the approach's success in fostering patient autonomy.

In conclusion, incorporating real-world examples of the 5P-Medicine approach in cardiovascular disease management substantiates its effectiveness. These examples reinforce the theoretical framework and provide concrete evidence of improved patient outcomes and empowerment. The shift towards a patient-centered, personalized, and proactive healthcare model promises a significant enhancement in the management of cardiovascular diseases.

## CONCLUSIONS

In summary, patient empowerment in healthcare is complex and dynamic, especially when developing medical devices and treatment plans for long-term illnesses like cardiovascular disease. A paradigm shift from the conventional compliance-oriented approach is represented by patient empowerment, which sees patients as active participants in their care and decision-makers with consequences. The problem is that empowerment has many different meanings and forms. Therefore, applying it consistently and successfully in healthcare requires a sophisticated understanding. Strategies for patient empowerment cover various topics, such as shared decision making, self-management, health literacy, and patient involvement. These tactics offer people, such as patients and healthcare professionals, the information, abilities, and encouragement they need to take charge of their health, make wise decisions, and actively participate in their healthcare process. To effectively empower patients to manage their CVD, shared decision making, patient involvement, health literacy programs, and self-management education are essential. By offering individualized, engaging, and readily available channels for patient involvement, integrating digital tools and technologies augments these approaches even more. Fostering collaborative and patient-centered healthcare approaches globally requires understanding the significance of patient empowerment and the customization of solutions to individual requirements, cultural contexts, and healthcare systems.

Encouraging individuals with CVD aligns with 5P-Medicine's tenets: Predictive, Preventive, Participatory, Personalized, and Precision Medicine. Giving CVD patients

access to machine learning and predictive analytics insights will enable them to recognize potential health hazards and take appropriate action. By encouraging patients to lead educated lifestyles and take proactive approaches to managing their health, the preventive component helps to delay the onset of chronic illnesses. The participatory component puts people at the center of healthcare choices by giving patients the resources and information they need to participate actively in their treatment. Furthermore, the tailored and precision components customize treatments for optimal efficacy by identifying unique genetic, environmental, and lifestyle characteristics. 5P-Medicine hopes to provide more accurate, individualized care by integrating these components. Achieving this goal requires patient empowerment, which promotes a patient-centered healthcare strategy that appreciates each person's preferences, experiences, and active involvement in their healthcare journey. This integrated approach improves the standard of care for patients with CVD and signifies a paradigm change toward a more thorough and customized approach to healthcare.

To achieve effective acceptance and a significant impact on healthcare, users' and patients' unique demands and preferences must be carefully considered during the design process of new medical devices. Wearables, artificial intelligence, decision support systems, signal processing, and data analysis are just a few of the many technologies that have been considered. These technologies all highlight how crucial user-centered design is to healthcare innovation. Customizing medical devices to meet each patient's needs is critical, particularly when managing cardiovascular disease. It promotes patient empowerment. User-centric design improves the usability, accessibility, and acceptance of medical technologies through various means, including clear user interfaces, tailored health insights, and robust data security protocols. Wearable technology and smart gadgets provide real-time monitoring and engagement opportunities, as the context of cardiovascular health has shown. Still, their efficacy depends on how well they integrate into users' daily lives. Patients' viewpoints and preferences can be given priority by designers to produce medical devices that satisfy clinical goals and are appealing to users. It will promote long-term usage and patient autonomy, increase patient empowerment, and improve health outcomes. Consequently, the successful adoption and meaningful integration of these technologies into the intricate healthcare ecosystem depend on a patient-centered approach to medical device design.

We present a thorough analysis of patient empowerment in CVD, stressing the value of a patient-centered strategy, incorporating digital tools, and adhering to the 5P-Medicine tenets. It also emphasizes that user-centered design is important when creating medical equipment for managing CVD. The questions that our findings raise highlight the need for more investigation and study in this area.

In response to the questions posed, the following actions are necessary: research and consensus-building within the medical community are essential for identifying and recognizing wearables and smart devices that satisfy clinical standards for diagnosis, monitoring, and therapeutic support in CVD; conducting surveys, interviews, and group discussions to gather insights on desired features, functionality, and outcomes related to CVD patient empowerment to understand the expectations of both patient and medical

communities; a thorough evaluation is required to ascertain whether current technologies live up to expectations. If not, defining the ideal scenario will require a collaborative effort between healthcare professionals, patients, and technologists to envision a comprehensive and effective ecosystem for CVD patient empowerment, taking into account factors like usability, accessibility, and integration into daily life; conducting a literature review will help identify existing studies, publications, and research papers that contribute valuable insights into solutions supporting diagnosis, monitoring, therapeutics, and self-management for CVD patients. Without these, identifying technological barriers and challenges will require thoroughly examining existing solutions, user feedback, and technological limitations.

The proposed study topics lay the groundwork for additional analysis and create a guide for patient empowerment with CVD. Subsequent investigations can further explore these questions, provide more detailed explanations, and advance the continuous development of patient-centered healthcare, especially with the empowerment of patients suffering from cardiovascular conditions.

### Funding

This work was supported by FCT–Fundação para a Ciência e Tecnologia, I.P. by project reference UIDB/50008/2020, and DOI identifier https://doi.org/10.54499/UIDB/50008/2020. This work is also supported by Fundação para a Ciência e Tecnologia under the project UIDB/00645/2020 (https://doi.org/10.54499/UIDB/00645/2020). Hanna Vitaliyivna Denysyuk is funded by the Portuguese Foundation for Science and Technology under the scholarship number 2021.06685.BD. The funders had no role in study design, data collection and analysis, decision to publish, or preparation of the manuscript.

### Grant Disclosures

The following grant information was disclosed by the authors:
FCT–Fundação para a Ciência e Tecnologia, I.P: UIDB/50008/2020.
FEDER-PT2020 Partnership Agreement: UIDB/50008/2020.
Fundação para a Ciência e Tecnologia: UIDB/00645/2020.
Portuguese Foundation for Science and Technology: 2021.06685.BD.

### Competing Interests

Ivan Miguel Pires is an Academic Editor for PeerJ Computer Science.

### Author Contributions

- Hanna V. Denysyuk conceived and designed the experiments, performed the experiments, analyzed the data, prepared figures and/or tables, authored or reviewed drafts of the article, and approved the final draft.
- Ivan Miguel Pires conceived and designed the experiments, performed the experiments, analyzed the data, prepared figures and/or tables, authored or reviewed drafts of the article, and approved the final draft.

- Nuno M. Garcia conceived and designed the experiments, performed the experiments, analyzed the data, prepared figures and/or tables, authored or reviewed drafts of the article, and approved the final draft.

## Data Availability

This work is a literature review and there is no raw data.

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
