# Peer review of "A roadmap for empowering cardiovascular disease patients: a 5P-Medicine approach and technological integration"

_PeerJ, doi:10.7717/peerj.17895_

## Round 0.1 · original submission · Major Revisions

Dear Authors,

Thank you for submitting your manuscript, titled "A Roadmap for Empowering Cardiovascular Disease Patients: A 5P Medicine Approach and Technological Integration," to PeerJ. The reviewers have carefully evaluated your manuscript, and based on their feedback, we have decided that your paper requires major revisions before it can be considered for publication.

Reviewer 1 noted that the manuscript is highly relevant and well-constructed but requires enhancements in statistical details, data accessibility, logical flow, and real-world examples. Specifically, the reviewer suggests providing a detailed description of the statistical methods, justifying the sample sizes used, and including visual representations of patient empowerment improvements. A clearer explanation of how each 5P Medicine principle contributes to patient empowerment and the inclusion of real-world examples would strengthen the manuscript.

Reviewer 2 commented that the review is well-written but recommended clarifying the target audience in the introduction, spelling out the term 'ICT' at first mention, and providing additional references for certain claims. Detailed information about the search methodology is also necessary.

Reviewer 3's comments are of particular importance. The reviewer observed that some content overlaps with a previously published article by the same authors and emphasized the need to clarify if this manuscript is a continuation of that work. They noted that the manuscript lacks sufficient empirical evidence, rigorous analysis, and a well-defined methodological approach, which makes it appear too theoretical or speculative. The manuscript's advocacy for a patient-centered approach and empowerment in cardiovascular disease management does not offer novel insights compared to existing work and may be viewed as an incremental contribution. It is essential to address these issues by providing more empirical evidence, a clearer methodological framework, and demonstrating how this work significantly advances current understanding or practice.

In light of these comments, we request that you thoroughly revise your manuscript to address the issues raised. Once revised, please submit the updated manuscript along with a detailed response to each of the reviewers' comments.

We look forward to receiving your revised manuscript.

Best regards,

Dr. Pedrino

Reviewer 1 ·

Basic reporting

The manuscrpt gives us a deep dive into how merging the 5P Medicine principles with tech could totally boost the way we help folks battling heart diseases. This topic is super relevant and seriously important, considering how many peeps worldwide are fighting these conditions. The paper's pretty well put together, offering a solid overview of what's already out there in the studies. But, to really up its game and make a mark in the field, it could do with some tweaks, especially in beefing up the stats details, making the data easier to get, smoothing out the logic flow, and backing up its points with some real-world proof.

Experimental design

1. It's kinda missing the nitty-gritty on the stats methods used to figure out if the strategies for making patients feel more in control are actually working. I'd say, throw in a bit under the methods section that lays out the stats tests, how sure we need to be to trust the results, and what counts as significant. This way, anyone trying to follow along can do their own check-ups.
2. Also, there's a bit of a hole where they should've talked about why they picked the number of people they did for their studies or data crunching. Bolstering the paper with some math to show the chosen sizes are enough to spot real differences or links would be solid .
3. The paper would really shine with some sharper visuals on how patient empowerment shifts with the 5P Medicine game plan. Maybe pop in a side-by-side graph or chart that makes it super clear how things get better for patients after these strategies roll out .

Validity of the findings

1. The leap from talking about 5P Medicine's core ideas to how they help empower patients feels a bit jumpy. Spelling out more clearly or maybe sketching a diagram that shows how each principle directly lifts up patients could smooth things over, making the study's thought process more straightforward
2. The wrap-up of the paper would hit harder with some straight-up examples from real life or studies that actually tried out the 5P approach with heart disease management. Dropping in a section that goes over how these ideas played out in the real world could really clinch the paper's arguments

Reviewer 2 ·

Basic reporting

The review article titled 'A Roadmap for Empowering Cardiovascular Disease Patients: A 5P Medicine Approach and Technological Integration' described possible ideas and approaches to support the empowerment of CVD patients. Overall, the review article is well-written. Some specific comments are listed as follows:
1. It would be good to clearly point out who will benefit from reading this review article in the introduction part, considering the scope of the review article. I assume not only healthcare providers will benefit.
2. In line 64, please spell out 'ICT' as this is the first time you mention it.
3. In lines 45-46, a reference is needed here.
4. In lines 52-54, a reference is needed here.
5. I notice that a lot of claims in the review, especially in the introduction, are supported by a single reference. As I understand, claims included in a review article may need more solid support across more than one study.

Experimental design

1. In lines 101-118, only the advancements of the search methodology are mentioned. However, I still do not know the details of the methodology and cannot evaluate it.

Validity of the findings

I have no comments here.

Reviewer 3 ·

Basic reporting

Some of the work presented seem to be already published in this article by the same set of authors. But they fail to clearly mention this is a continuation of their work that was published earlier.

https://www.ncbi.nlm.nih.gov/pmc/articles/PMC8587644/

Experimental design

Despite the conceptual merits if stated as a continuation of previous work, the paper lacks sufficient empirical evidence, rigorous analysis, or a well-defined methodological approach to support the proposed patient empowerment strategies and integration of digital tools, it may be considered too theoretical or speculative in nature.

Validity of the findings

The advocacy for a patient-centered approach and the conceptualization of empowerment in CVD management is not substantively different from existing work in the field, and the paper does not offer novel insights or a unique perspective, it may be viewed as an incremental contribution that does not significantly advance the current understanding or practice.

---

## Round 0.2 · accepted · Accept

Dear Authors,

Thank you for submitting your manuscript titled "A Roadmap for Empowering Cardiovascular Disease Patients: A 5P Medicine Approach and Technological Integration" to PeerJ. We are pleased to inform you that, after a thorough review process and considering the feedback provided by our reviewers, your revised manuscript has successfully addressed the primary concerns raised.

The reviewers have confirmed that the modifications made in response to their comments have significantly improved the quality of the manuscript. Therefore, we are delighted to inform you that your manuscript has been accepted for publication. We commend you for the diligent work and the improvements made based on the reviewers' feedback.

Thank you for choosing PeerJ to publish your important research findings. We look forward to your future contributions.

Kind regards,

Dr. Pedrino

Reviewer 1 ·

Basic reporting

no

Experimental design

no

Validity of the findings

no

Reviewer 2 ·

Basic reporting

All my concerns have been addressed.

Experimental design

All my concerns have been addressed.

Validity of the findings

All my concerns have been addressed.